# Quantifying Mineral-Ligand Structural Similarities: Bridging the Geological World of Minerals with the Biological World of Enzymes

**DOI:** 10.3390/life10120338

**Published:** 2020-12-10

**Authors:** Daniel Zhao, Stuart Bartlett, Yuk L. Yung

**Affiliations:** 1Division of Geological and Planetary Sciences, California Institute of Technology, Pasadena, CA 91125, USA; dxczhao20@gmail.com (D.Z.); yly@gps.caltech.edu (Y.L.Y.); 2Department of Mathematics, Harvard University, Massachusetts Hall, Cambridge, MA 02138, USA; 3Earth-Life Science Institute, Tokyo Institute of Technology, Tokyo 152-8550, Japan; 4NASA Jet Propulsion Laboratory, Oak Grove Dr, La Cañada Flintridge, CA 91011, USA

**Keywords:** origin of life, minerals, enzymes, ligand, astrobiology

## Abstract

Metal compounds abundant on Early Earth are thought to play an important role in the origins of life. Certain iron-sulfur minerals for example, are proposed to have served as primitive metalloenzyme cofactors due to their ability to catalyze organic synthesis processes and facilitate electron transfer reactions. An inherent difficulty with studying the catalytic potential of many metal compounds is the wide range of data and parameters to consider when searching for individual minerals and ligands of interest. Detecting mineral-ligand pairs that are structurally analogous enables more relevant selections of data to study, since structural affinity is a key indicator of comparable catalytic function. However, current structure-oriented approaches tend to be subjective and localized, and do not quantify observations or compare them with other potential targets. Here, we present a mathematical approach that compares structural similarities between various minerals and ligands using molecular similarity metrics. We use an iterative substructure search in the crystal lattice, paired with benchmark structural similarity methods. This structural comparison may be considered as a first stage in a more advanced analysis tool that will include a range of chemical and physical factors when computing mineral-ligand similarity. This approach will seek relationships between the mineral and enzyme worlds, with applications to the origins of life, ecology, catalysis, and astrobiology.

## 1. Introduction

Life and its environment, the biosphere and geosphere, have always been inextricably linked. Despite the epistemological separation between organisms and their environs, an understanding of life is fundamentally incomplete without an understanding of the non-living aspects of its surroundings, with all the myriad interactions entailed therewith. The vast majority of known life resides within or upon Earth’s geosphere, which provides physical structure, chemical free energy, and material substrates, without which life as we know it would be impossible.

There are myriad feedbacks between life and the geosphere. While we often view the physical environment as providing evolutionary challenges and constraints upon which organisms can persist, life’s existence and chemical byproducts also shape which minerals, rocks and gases are most readily produced. Hence it is impossible to study life out of its environmental context, and any analysis of geospheric evolution is futile if life’s influence is overlooked. Enzymes are arguably the most sophisticated and versatile inventions of biological learning and evolution, which have enabled the colonization of a vast range of niches, and can facilitate nanoscale chemical and physical processes that are far beyond the reach of human engineering. Enzymes are primarily composed of amino acids, a relatively small but apparently universal set of 20 organic compounds. However, enzymes rely upon inorganic components for their function, particularly active site catalytic power. These inorganic molecules or clusters, shape and are shaped by the geosphere and abiotic environment. Their raw components must be sourced and sequestered from the environment, but their catalytic action is often aimed at chemically altering or metabolizing other compounds from the environment. The byproducts of these processes in turn feed back on the chemical and physical conditions of the environment.

A prime example is chemolithoautotrophic organisms, which exploit reduced mineral compounds as a source of free energy for metabolism. However, these minerals are not just consumed by such organisms, their own chemical evolution is also strongly affected. In the case of ferrous iron [Fe(II)]-oxidizing bacteria, chemical weathering has been shown to be enhanced due to the presence of such bacteria. In fact, it was found that “chemolithotrophic Fe(II)-oxidizing bacteria are likely contributors in the transformation of rock to regolith” [1,2].

In addition to the coevolution of life and environment over the last 3.5 billion years, the abiotic conditions at the time of the origins of life played a fundamental role in fostering and constraining those crucial events. Mineral compounds have been highlighted as candidate materials for catalysis [3,4,5,6,7,8,9,10,11], information storage [12,13,14], and compartmentalization, concentration and polymerization [5,11,15,16,17], among others [4,5,13,18]. Focusing on catalysis, many authors have highlighted the compelling structural similarities between key inorganic enzyme clusters, and minerals [7,8,19,20,21,22,23,24,25,26]. Iron-sulfur clusters are employed across a vast range of essential biochemical functions [27,28]. For this reason, significant efforts have explored the potential connections between minerals, and the enzyme families of ferredoxins, nitrogenases, NADH dehydrogenase, and CO-dehydrogenase, among others.

While these structural analogies are highly compelling, there is a relative dearth of systematic, quantitative comparisons between the structures of minerals and metalloenzyme clusters. Understanding the origins of life requires a simultaneous application of constraints from the ancient geological world and the known biological world. As such, it is essential to systematically assess how biological machinery and geological compounds are related. Structurally, we must quantify the similarities between inorganic enzyme active sites and minerals, to ascertain whether biochemical catalysts are truly derived from mineral analogues. Functionally, we must establish whether minerals can function as catalysts, and for those that are weakly catalytic, predict the modifications (by amino acid addition or otherwise) required to achieve catalytic efficacy.

The present work was motivated by these questions. We aimed to lay an analytical foundation for the large scale comparison of minerals and metalloenzyme ligands. In this context, we refer to “ligands” as the active centers of metalloenzymes often involved in catalytic activity. Given the unprecedented recent advances in chemoinformatics, rational ligand design and big data [29,30], methods for mineral-enzyme comparison applied to large datasets have vast potential to extract hitherto unknown connections between the geological and biological worlds, predict ancient catalytic compounds relevant to life’s origins, and could even help constrain life-environment relations on other planets. It could also reveal mineral compounds with functional potential that had previously not been considered.

In this paper we report on a first set of mineral-enzyme structural comparisons using a mature chemoinformatic toolbox known as RDKit [31,32]. We sourced mineral structures from the Crystallography Open Database [33], and enzyme data from the Protein Data Bank [34,35]. In this initial phase, the focus was Iron-Sulfur and Nickel-Iron-Sulfur ligands, since these play such a crucial role in biochemistry and possibly its origins. Comparisons with a range of iron-sulfur mineral structures confirmed many previous structural analogies but also highlighted some negative results. We propose the presented methodology as a powerful technique that can now be developed to include chemical characteristics, and applied to large mineral and enzyme databases to build networks of relationships between the geological and biological worlds.

Section 2 presents the main aspects of our methodology, including mineral structure extraction and similarity index computation. Section 3 presents our similarity index results and compares them with previous speculations on mineral-enzyme structural analogies. We conclude in Section 4.

## 2. Methodology

This work aims to establish the groundwork of a method to objectively compare mineral and ligand structures. A unique challenge with making mathematical comparisons between the structures of mineral-ligand pairs, as opposed to traditional molecular similarity comparisons, is the arrangement of atoms in the mineral crystal lattice, which does not define a single molecule for comparison. Unit cells are often used as a basis for visual comparison between minerals and ligands that share alike structural motifs. However, establishing a mathematical index of similarity requires a more methodical approach, as the scope and size of the unit cell is variable and oftentimes incongruous. The aim of our method is to overcome this problem by extracting a molecular substructure of similar size to the target ligand from the crystal lattice, in order to make an analogous comparison with the ligand.

Using crystallographic data available on the Open Crystallography Database, we generated the structural coordinates of supercells of various minerals. To estimate the size of the ligands, we constructed an oriented bounding box around the target ligand using its structural coordinates found on the Protein Data Bank. The dimensions of the bounding box were normalized to the average bond length of the ligand and resized to the coordinate space of the mineral lattice. We iterated the bounding box through several increments of axis rotations and translations across the lattice, selecting all atoms and bonds within the bounding box as the molecular substructure for that iteration. Once a substructure was selected, we calculated its molecular similarity index with the ligand.

Over several transformation iterations, the index with the largest value and its respective substructure was retained as the final similarity index, as depicted in Figure 1.

The similarity indices were calculated using RDKit-generated molecular fingerprints [36] and the Sorensen-Dice similarity coefficient. RDKit fingerprints are topological representations of molecular structure generated by hashing all subgraphs within a specified range of path sizes into bit IDs. Many fingerprinting algorithms exist, but we chose RDKit’s implementation for its versatility in dealing with non-aromatic molecules and its relatively low computation time, since many nested iterations of substructure search require significantly longer running times for most 3D fingerprinting methods.

For the same reason, Dice similarity [37,38] was chosen as an effective and quick metric to quantify similarity. The usefulness of Dice similarity and comparable mathematical metrics have been documented in several studies from a wide range of fields, including chemoinformatics [39,40] and medicine [41,42]. Deferring to standard chemoinformatic practices [43,44], we used a similarity index of 0.8 as a rough threshold for inferring two molecules were structurally correlated, while we considered an index of 0.65–0.8 as suggestive of a potential correlation. However, as explored in Baldi and Nasr [45], the usefulness of such a threshold depends largely on the dataset. Especially in this context, the “significant” range of similarity is defined essentially for qualitative use since mineral compositions and relative ligand sizes can vary greatly.

## 3. Results

Most studies on catalysis at the origins of life have focused on a select group of ligands involved with key metabolic pathways, and their potential mineral predecessors. For example, Russell and Hall [25] proposed that nickel-substituted greigite served as a catalytic precursor to the active C-cluster of acetyl-CoA synthase (ACS/CODH) and potentially other metalloenzyme cofactors, noting its structural similarity as suggestive of similar functionality. In Wächtershäuser’s “iron-sulfur world” hypothesis, ferrous surfaces near hydrothermal vents play a pivotal role in primitive metabolic chemistry, specifically in electron transport and catalysis [10]. Notably, it was also proposed that the first iron-sulfur clusters arose from pyrite-like surfaces and were partially retained as ligands in ancient ferredoxins, suggesting a structural relatedness between the two. We now know that pyrite has very limited catalytic ability, but in the last two decades interest has shifted toward iron sulfide minerals [7,19] and iron hydroxides [3,20,46].

Given the aforementioned importance of iron sulfide clusters, significant work has aimed at synthetically bridging the gap between iron sulfide minerals and simple enzymes, both in the origins field [21,26,47,48,49] and chemical engineering [50,51]. Ferredoxins are considered one of the most ancient protein structures in the biosphere [52,53,54], and make use of the $\left[Fe_{2}S_{2}\right]$, $\left[Fe_{3}S_{4}\right]$, and $\left[Fe_{4}S_{4}\right]$ clusters. In the following, all three of these structures were compared with key minerals that have been propounded as prebiotic catalysts. Using our described method, we generated a similarity index matrix of relevant ligands and metal-sulfide minerals, presented in Figure 2.

As expected, mackinawite and greigite share many structural similarities with the iron-sulfur clusters. This structural affinity has been documented in multiple studies [23,25], and Russell and Hall [24] proposed that a protoferredoxin cluster could have arisen in an intermediate oxidation state between mackinawite and greigite. The alikeness of both minerals towards the $\left[Fe_{2}S_{2}\right]$ and $\left[Fe_{4}S_{4}\right]$ clusters supports this hypothesis. The similarity indices of unity (perfect match) imply structural indistinguishability, but notably not functional equivalence. While some structural resemblance is expected in mineral-ligand pairs of similar function, the converse is not true, as molecular structure alone may not predict chemical bonding behavior.

The less well-studied minerals marcasite and troilite demonstrate an unexpectedly high structural affinity with iron-sulfur clusters as well, providing a new avenue to study the poorly-understood bridge between minerals and metalloenzyme clusters. Though less abundant than pyrite, these minerals still compose a significant portion of metal sulfides in locality counts [7]. This example highlights the utility of our method to reveal less studied minerals that could be more important to early metabolic catalysis than previously thought.

Another noticeable result is that most ligands have relatively low similarity indices with pyrite and chalcopyrite compared to the other iron-sulfur minerals, supporting the notion that pyrite surfaces are functionally disparate from iron-containing clusters, and had suboptimal catalytic properties for early metabolic chemistry. One explanation for this is provided by Russell and Hall [24], who argued that pyrite’s tetragonal crystal structure inhibited nucleation and reduction–common features of iron-sulfide clusters in metalloenzymes. The preferred hypothesis of Krupp [55] involves the oxidation of mackinawite to greigite facilitated through electron conduction, motivated in part by their closely related crystal structure. Unlike the catalytically inert pyrite, the oxidized greigite can participate further in protometabolism.

It is noted that nickel-containing ligands and minerals show significantly lower similarities relative to their pure iron-sulfur counterparts. This is likely partially an artefact of the underlying similarity method. Unlike many network-based 3D fingerprinting methods, the RDKit fingerprint is based on atom and bond arrangements and does not consider the spatial geometry of the molecule. It is therefore especially sensitive to differences in atom types, which explains the large drop in similarity index when iron is substituted with nickel. With respect to its catalytic properties, as shown by Cody et al. [19] and Huber et al. [56], these nickel-containing minerals exhibit significantly different behaviors in organic chemistry (ie. enhancement of the Koch reaction) compared to iron-sulfur minerals, and are perhaps involved in different metabolic pathways. The substitution of a non-iron transition metal atom could drastically alter its catalytic potential, so the observed drop in similarity due to nickel-iron mismatches is not necessarily detrimental from a functional perspective. Regardless, we acknowledge this potential bias of the similarity algorithm and stress the importance of examining chemical and electronic properties of the minerals in conjunction with molecular structure.

To re-examine past structural comparisons through a computational lens, we applied our method to a selection of mineral-ligand pairs that were compared visually in previous literature. These are shown in Figure 3.

The $\left[Fe_{4}S_{4}\right]$ cluster exhibits both high visual and molecular similarity with the cuboid structure of greigite. Likewise, as shown in (c), the visual comparison made in Nitschke et al. [8] between the CODH/ACS C-cluster and nickel-substituted greigite is also supported by its high similarity index. Both these factors raise the intriguing possibility that there could exist an intimate relationship between the two that is functional and evolutionary in nature.

Adversely, the structural disparity between the $\left[Fe_{7}MoS_{9}\right]$ cluster and the greigite substructure extracted in (e) seems to indicate a much weaker or indirect potential relationship, though the cuboid motif appears to be shared [25,57]. Other comparisons, such as those shown in (b) and (d), are visually compelling but have relatively low indices, suggesting that the actual structural affinity of the pair may be less than the apparent aesthetic match. This demonstrates the advantage of a quantitative metric like “Similarity Index” over intuition.

## 4. Conclusions

Our study presents a foundational, mathematical approach to comparing mineral-ligand pairs relevant to the origins of life and to the large scale structural comparison of contemporary ligands and minerals. Visual comparisons are useful but limited in scope, and are not always reliable. Screening of large geological data for structural similarities kindles opportunities to identify new, potentially important minerals, and diversifies the existing pool of metal sulfides to investigate in a guided manner. A significant majority of minerals have yet to be studied–not because of their irrelevance, but rather due to the difficulty of choosing the right minerals and ligands to investigate. By harnessing cutting-edge chemoinformatics and drug discovery research in origins of life studies, we provide a powerful angle for exploring the myriad connections between minerals and enzymes.

However, the present study is a starting point with the aim of building a more comprehensive comparison tool that will include chemical factors such as redox potential, charge, magnetic state, kinetic properties and catalytic efficacy. Structural similarity is an important but clearly insufficient metric given the complex array of genotype-structure mappings found in the biological repertoire (structurally similar ligands that are phylogenetically distant and vice versa). Ultimately, the presented methodology may serve as a coarse-grained filter phase, feeding into further steps of chemical and physical comparison. This more comprehensive tool will be able to identify mineral-ligand pairs that are structurally similar but in fact share little to no chemical or catalytic similarity (which may be the case for some of the structurally similar pairs highlighted in this work).

Astrobiology as a discipline has made remarkable advancements in many diverse branches of thought, including bottom-up laboratory experiments on mineral catalysis, top-down phylogenetic analyses, and topological studies of enzyme structure. We believe a data-driven, informatic approach to the field may help unite these many fragmented ideas and, in the mineral context, compartmentalize the geological world into a new system of groups as a basis for study. Classifying minerals based on their inherent catalytic abilities, functional similarity with biomolecules, or association with peptides would enable a systematic means of expanding the field and closing the gap between bottom-up geological insights, and top-down biological and phylogenetic studies. The approach presented herein could readily be expanded to fulfill this grand objective.

## Figures and Tables

**Figure 1 life-10-00338-f001:**
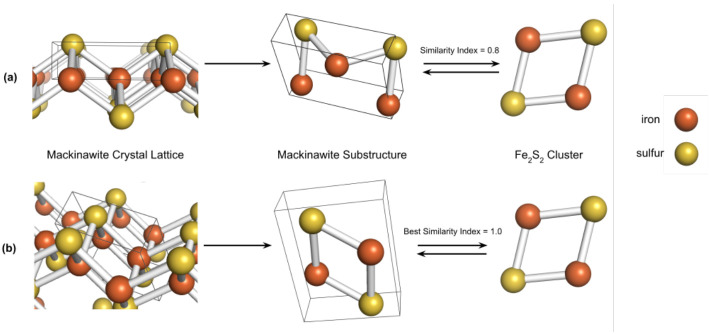
Example of substructure search in the crystal lattice of mackinawite, using the $Fe_{2}S_{2}$ cluster as the target ligand. Shown in (**a**), an incongruent substructure which leads to a sub-optimal similarity index, while the substructure encompassed in (**b**) is chemically identical to the ligand.

**Figure 2 life-10-00338-f002:**
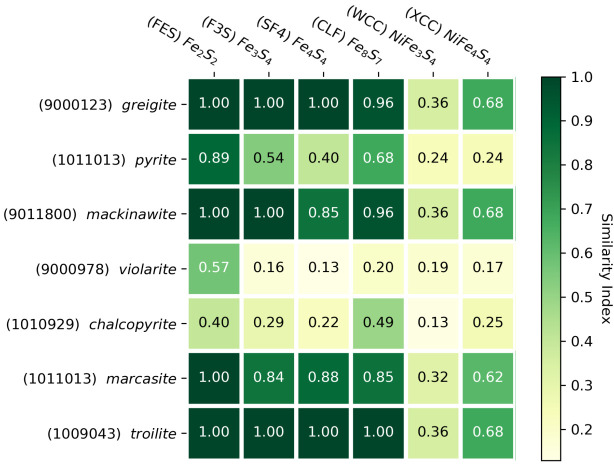
Heatmap of similarity coefficients between selected minerals and ligands with associated COD and PDB ligand identifiers.

**Figure 3 life-10-00338-f003:**
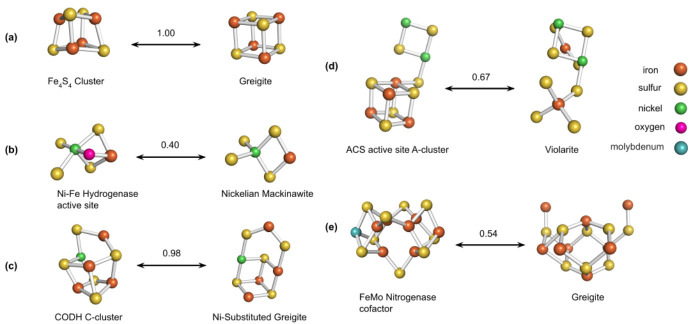
Structural comparisons using similarity index (number above arrows) of (**a**) $\left[Fe_{4}S_{4}\right]$ and greigite, (**b**–**d**) nickel-containing ligands and minerals from Nitschke et al. [8], (**e**) $\left[Fe_{7}MoS_{9}\right]$ nitrogenase cofactor and greigite [25].

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
