# Peer review of "Quantifying Mineral-Ligand Structural Similarities: Bridging the Geological World of Minerals with the Biological World of Enzymes"

_life, 2020, doi:10.3390/life10120338_

Round 1

Reviewer 1 Report

i) This manuscript present a mathematical approach to detect comparable structure in minerals and enzymes. As authors said finding the “good” mineral from the structure of metalloenzyme, to understand the “origin of life”, is particularly interesting. However structure of cluster is an interesting starting point but not the key point. In biology reactivity is dictated by the redox potential of metal cluster and having the same structure will not always prove that they are doing same catalysis. Actually finding redox potential is common and usually done in enzymes but for mineral, which contain several metals, the study is more difficult. From my perspective, it would be important to notice in the manuscript that another key point is to determine the redox potential and combining structural comparison with redox potential could really be powerful.

ii) In the same idea, authors explain lines 147 to 149 that “the structural dissimilarities apparent between iron-sulfur clusters and the pyrites suggests that ancient ligands were more likely derived from other iron-sulfur minerals such as greigite that may have been less abundant but more structurally similar”, I would be less confident about this affirmation as it’s true in most cases but in biology, we can find enzymes structurally close but phylogenetically uncorrelated, and at the opposite phylogenetically close but structurally different. I agree that comparing structures of mineral vs enzyme with computational method could give new ways that have to be explore but it has to be combined with other fields like geochemistry, bioenergetics and physics to avoid any wrong conclusion. For example you said that visual comparisons are not always reliable which is true, but for example in the Nitschke’s paper, the comparison has been made because geological data of hydrothermal vent and phylogenetic data were converging in the direction of a common catalytic role of NiFe hydrogenase and Mackinawite, so in my opinion it is not just an intuition but it is based on data.

iii) Concerning the method, you never explain what is for you, a good or a bad similarity in your “range of similarity”. It would be interesting to explain for each range if the similarity is good or not and if you consider that is good enough to conclude that they are structurally correlated.

iv) I also suggest to the author to change the term “ligand” by “active center”. In structural biology and particularly for metalloprotein, the term ligand is used to describe the amino acids ligating the metal cluster and that have in most case a role in changing the metal catalytic properties.

v) Concerning the figure, I suggest to add a legend in figure 1 (red for iron and yellow for sulfur) and it would be helpful to have the pdb number of the proteins and the COD number of the minerals.

vi) for taping corrections: line 54 “CO-dehydrogenase”, line 57 “structures”, line 130 “expected”

Conclusion:

This manuscript is an interesting study to quantify similarities between structure of mineral and cluster of enzyme. It can help to find new minerals to explore and I liked the fact that a method can give a direct relationship between the mineral and enzymes world. I suggest to make some changes in the manuscript in order to explain that, there is a step to fill the gap between the "bottom-up" (theories of origin predicting the transition from the non-living to the living) and the "top-down" (reconstruction of the founding system based on our knowledge of catalysis in extant metallo-enzymes) that has to be fulfill by different methods and fields. Computational study is one of them and combined with biological, geochemical and biophysical studies, could lead to a better understanding of origin of life.

Author Response

We thank the reviewer for their insightful and useful comments and address them sequentially below:

i) We completely agree that the redox potential is an important factor in a comparison such as that proposed in our paper. While we did not include this in the present work, we anticipate this being the next phase in the project. This has now been stressed in the new paragraph in the conclusions.

ii) We agree with these points and have thus added text (see new text near the end of results section) and references to more detailed explanations in literature, and made changes to the appropriate sections to better reflect our goal of emphasizing the foundational characteristic of the work. In particular, as the reviewer has constructively pointed out here and the above comment, we stress the importance of considering a wider range of factors including the chemistry and redox potential of minerals, and clarify that molecular similarity is useful but not sufficient.

iii) This is an interesting point we had considered during the analysis. Using chemoinformatic standards of a “good” similarity range, we considered anything above 0.8 similarity as strong evidence of structural correlation and 0.65-0.8 as potentially indicative of a correlation. However, we were hesitant to use this definition too much in the paper because similarity thresholds are context-dependent (see Baldi 2011, “When is Chemical Similarity Significant?”), so it is difficult to predict a “good” similarity range given a new dataset and our small sample size. Nonetheless, we believe this is an important question that could be answered in the bigger project of creating an informatic approach to astrobiology. These considerations have been added to the methodology section.

iv) We agree that active center is an appropriate term to use. However, we feel that replacing all instances of ligand with active center would disrupt the flow of the text. Therefore, we have added a phrase in lines 68-69 to clarify our use of the phrase ligand.

v) A legend has been added to figure 1, and PDB ligand identifiers and COD numbers have been added to figure 2.

vi) These have been corrected

Conclusion: we have augmented the discussion in the conclusion of the astrobiological relevance of this work regarding the connection between bottom-up and top-down approaches

Reviewer 2 Report

Reviewer comments:

Title: Quantifying Mineral-ligand Structural Similarities: Bridgingthe Geological World of Minerals with the Biological World of Enzymes

Authors: Zhao, Bartlett, Yung

Synopsis: The question of how much of metal-mediated catalysis by enzymes is dictated by metal-ligand chemistry rather than by complexities available only through evolution, and by extension what continuity may exist across the span from minerals to enzymes, is now a mainstream problem in Origin of Life research. There seems to be a body of thorough chemical research on the design of analogues to protein metal cores in the bioinorganic chemistry literature, but somewhat less that has been extended to the search for mineral analogues. The current paper takes early steps in filling that need with systematic methods, addressing the stages of combinatorial search for similar bond topologies within mineral unit cells to templates from protein metal cores, and quantification of the degree of overlapin bond topology between the template and its match in the mineral. The combinatorial search stage, and graph isomorphism resolver, are programmed by the authors. The topological similarity index is derived from a Novartis tool (RDKit) used to compare molecule fragments for pharmaceuticals, and a very simple binary vector-overlap metric (Sorensen-Dice) is applied to RDKit fingerprints as a measure of similarity. The algorithm was chosen to prioritize speed and simplicity; thus it does not employ bond distances or angles, or any other chemical property besides atomic species at the graph vertices.

This is a compact paper. It only aims to address a small part of the problem of mapping protein metal cores to mineral cluster analogues. The methods chosen to do that seem reasonable and the implementation looks believable. Because visual inspection and rather ad hoc suggestions of mineral analogy to biological metal cores are now quite widespread in the Origins literature, I welcome efforts to search more thoroughly and to automatically rank hypotheses of this kind. At the same time, it is well appreciated within both molecular biology and bioinorganic chemistry that the catalytic capacity of a metal core depends on a large collection of factors: charge, magnetic state, midpoint potentials, and exposed HOMO/LUMO orbital geometries, all of which generally depend on bond angles, lengths, and bonded species. These dependencies can be quite sensitive for strained configurations or labile valences. So I see the place of the current work as a very coarse pre-processing filter, meant to handle problems of combinatorial search and reduce the number of possibilities that must then be considered with much more costly methods.

Recommendation: After a lot of uncertainty and waffling, I am willing to recommend the paper for publication more or less as it is. My hesitation has been that this topology mapping addresses a very small part of the problem, and I expect that experts who work on the chemistry in this area may question what can be concluded from the results of this stage in isolation.

I reached out to my colleague (without mentioning this particular review) to understand the areas for caution, because I know he has done work comparing the Fe2S2 motif in mackinawite to one of the enzyme centers, and shown that the transitions in the two and thus their catalytic roles are completely dissimilar. He says that work is still in prep, and I apologize that I cannot give the authors a source, but he referred me to the following review as a reflection of reasonably modern analysis in the bioinorganic chemistry literature:

Venkateswara Rao and R. H. Holm Synthetic Analogues of the Active Sites of Iron-Sulfur Proteins Chem. Rev. 2004, 104, 2, 527-560

https://doi.org/10.1021/cr020615+There is some commentary by the authors throughout Sec. 5.6, 5.6, 6, and 8, and extensive data, showing what can be modulated.

I guess I submit for the authors' consideration the following difficulties in using a unit-cell analysis to do much more than create a pre-process filter:

  1. A bond pattern that can be matched within a unit cell is not reflective of the whole unit cell or of the protein environment to which it is being compared. The unit cell or protein environment also consist of all the other bonds that complete the valences of the atoms. In proteins or bioinorganic metal-organic analogues, the charge, magnetic state, midpoint potentials, and exposed HOMO/LUMO orbital geometries can be somewhat sensitive (even quite sensitive, for strained configurations or labile valences) to bond angles, lengths, and bonded species. Whether a cluster is catalytically comparable turns on these features, as much as on the overtly similar topology.
  2. It is probably true by construction that all catalysis on minerals is done at surfaces, and much of it probably at defects or impurity centers. Neither of those has the same bond properties as the unit cell in the interior. It may actually be a strength of this paper's partial-match approach, that cuts through a unit cell are proxies for surface patterns, but the fact that bonds to the solvent must complete valences is likely a source of even greater interpretive difficulty than the comparison of interior to surface motifs.

I won't place a condition that the authors must address these points, which would go outside the scope of work they have done. But a compact, informed, discussion that raises the reader's awareness of what part of the problem the current algorithm addresses, and how much else stands between that filter and the interpretation of catalytic meaning, it would make the paper a stronger and more valuable contribution.

The authors might also run their document through a spell-checker, and give a slow proofread. There were unusually many simple typos that could have been caught.

Author Response

We thank the reviewer for their insightful and useful comments. We have added text to emphasise that our approach is the first step in a larger endeavour that will eventually include chemical and physical properties, including several that the reviewer suggested. We have also suggested, as the reviewer recommended, that the presented methodology may serve as an initial filtering phase within a large comparison tool (see conclusion section).
We have added the Rao 2004 reference to the introduction and corrected typographical errors.

Reviewer 3 Report

I would like to start my review by highlighting the sentence on line 56 "While these structural analogies are highly compelling ...". The thesis of my review is that the structural analogies discussed in literature are NOT compelling as presented here.

I appreciate the mathematical analysis of a structural "appearance" model that takes into account the connectivity and spatial arrangement, but only in part. Critically, the model falls short of fundamental chemistry of metal-ligand interactions. Combinatorial approaches such as "unprecedented recent advances in chemoinformatics, rational ligand design and big data" into structure/function relationships are misleading and incomplete without considering electronic structural features such as bonding and back-bonding, covalent, ionic, and magnetic interactions. The latter is particularly important for Fe-S clusters.

When considering the chemical bonding principles, the similarity index between the mackinawite substructure and the [2Fe-2S] cluster of biology and biomimetic chemistry is ZERO!
1) The Fe ions in mackinawite is coordinated by four mu-4 bridging sulfides. The Fe ions in plant ferredoxins is coordinated by two terminal (kappa-1) thiolates and two mu-2 bridging sulfides.
2) The (FeS)2 units in mackinawite is part of an extended material with a band structure. The [2Fe-2S] cluster in ferredoxins is a localized, molecular unit.
3) The Fe-S-Fe/Fe...Fe magnetic interaction in mackinawite in a periodic lattice unit has little similarities to the localized magnetic interactions in ferredoxin clusters.
4) and the list of critical differences goes on with different Coulomb interactions, pi/pi stacking and lack of, etc.

The dissimilarity of "iron-sulfur and nickel-iron-sulfur ligands" in nickel-substituted greigite and the "catalytic precursor to the active C-cluster of acetyl-CoA synthase" is even more striking according to chemical bonding and coordination chemistry principles.

Thus, according to the above interpretation of similarity/dissimilarity among minerals and biological clusters, Figure 2 (while mathematically correct according to the structural "appearance" model) is misleading and puts emphasis on the incorrect locus of minerals, which is bulk structure.

Instead, we all (as the community) must redirect our focus on surface formations of minerals (edges, sheets, apexes, grain boundaries). These are not just dependent from the chemical environment (lithosphere, atmosphere, hydrosphere), but also greatly different from the bulk structure.

The reason I signed my review is to offer my collaboration to expand the authors' strong mathematical approach to include chemistry concepts such as inner and outer-sphere coordination environments (at least).
I wish to emphasize that my lack of support is not because of the authors work and effort. However, I cannot stand behind the central idea of the work presented in the manuscript that can be refuted with a few 'quick-'n-simple" quantum chemical calculations. I do not wish citations to the authors' work showing why the discussed similarities are incorrect.

Author Response

We thank the reviewer for their insightful and useful comments. Indeed we acknowledge that structural similarity is a necessary but clearly insufficient metric for enzyme-ligand chemical comparison. We have scaled back the language of the text in key places in order to emphasise that this is the first phase of a larger endeavour that will incorporate chemical and physical characteristics on top of the structural aspect considered here. We appreciate the concern about negative citations, hence we have tried to stress in the text that this is a structure-only comparison and hence acknowledge the existence of false comparisons that could arise from the approach as it stands now. In the abstract and conclusions we have stated that this phase will be included as one level in a multi-level approach that will include chemical and physical properties. We hope that this alleviates the risk of negative citations since we are not claiming that our current approach includes all the relevant factors.
We very much appreciate the offer of collaboration and look forward to this.

Reviewer 4 Report

The present article written by Zhao et al. provides a mathematical method in bridging the connection between minerals and enzymes. This seems like an interesting idea, but authors did not investigate this idea thoroughly. This referee feels the authors should address these points in detail as shown below in order to publish in this journal.

  1. The authors claimed that the methodology can be expanded and applied to large mineral and enzyme databases. While the method is able to describe the relation between iron-sulfur minerals and ligands to some extent, it does not hold good for nickel-containing minerals and ligands. The authors explained that their method is sensitive to difference in atom types. In this background the applicability of this method for general mineral-enzyme pairs is questionable. I would suggest the authors to revise their claim accordingly throughout the manuscript.

  1. The authors claim on the implication of the method is very much confusing throughout the manuscript. In the abstract they mentioned that their method is based on structural and chemical similarity. However, the title of the paper reads “…. structural similarities”. Moreover, Line 134 says “the similarity indices of unity (perfect match) imply structural indistinguishability”. However, in line 166-168 authors explain their observations is based on chemical and structural affinity of the mineral-ligand pairs.

  1. The manuscript lacks proper references of the methodology used like RDKit fingerprint, Sorensen-Dice similarity etc.

  1. The authors claim that ancient ligands likely derived from leas abundant mineral greigite instead of more abundant pyrites. Can they suggest any reference supporting this statement?

  1. In figure 3 the color of iron and oxygen atom is very much confusing.

Author Response

We thank the reviewer for their insightful and useful comments and address them sequentially below:

1. We acknowledge that the RDKit method’s sensitivity to atom types may be a shortcoming of the method, but we do not believe this invalidates its usefulness in general application. Since metallic substitutions for iron in minerals and ligands can significantly alter their catalytic role in metabolic pathways, the lower similarity resulting from nickel-iron mismatches is not completely unexpected. The section addressing this phenomenon has been expanded upon to recognize these potential biases, and we have qualified the central claim by emphasizing the key importance of considering bond chemistry and electronic properties. Our paper primarily seeks to lay some groundwork on an informatic approach to astrobiology, and not to draw definitive conclusions on structural correlations.

2. The “chemical similarity” as it is used here is the same as structural and molecular similarity, specifically the RDKit fingerprint and Sorensen-Dice coefficient similarity metric. Its redundant/confusing uses have been removed and replaced with “molecular similarity”.

3. References to RDKit and Sorensen-Dice similarity have been added.

4. The implication of this claim was not intended to be as strong as it was, we have replaced this claim with a reference to more detailed explanations in literature.

5. The oxygen has been recolored in figure 3.

Round 2

Reviewer 4 Report

The authors have addressed my concern carefully. I recommend publication of the manuscript.